# Acoustic Emission Monitoring of Fatigue Crack Growth in Hadfield Steel

**DOI:** 10.3390/s23146561

**Published:** 2023-07-20

**Authors:** Shengrun Shi, Guiyi Wu, Hui Chen, Shuyan Zhang

**Affiliations:** 1Centre of Excellence for Advanced Materials, Dongguan 523808, China; guiyi.wu@ceamat.com (G.W.); shuyan.zhang@ceamat.com (S.Z.); 2School of Materials Science and Engineering, Southwest Jiaotong University, Chengdu 610032, China; xnrpt@swjtu.edu.cn

**Keywords:** crack growth, acoustic emission, Hadfield steel, power law

## Abstract

Evaluating the condition of a Hadfield steel crossing nose using existing inspection methods is subject to accessibility and geographical constraints. Thus, the use of conditional monitoring techniques to complement the existing inspection methods has become increasingly necessary. This paper focuses on the study of acoustic emission (AE) behaviour and its correlation with fatigue crack growth in Hadfield steel during bending fatigue tests. The probability density function for acoustic emission parameters was analysed based on the power law distribution. The results show that a sharp increase in the moving average and cumulative sum of the AE parameter can give early warning against the final failure of Hadfield steel. Two parts (Part 1 and Part 2) can be identified using the change in the slope of duration rate (*dD*/*dN*) vs. Δ*K* plot during the stable fatigue crack growth (FCG) process where Paris’s law is valid. The fitted power law exponent of AE parameters is smaller in Part 2 than in Part 1. The novelty of this research lies in the use of the fitted power law distribution of AE parameters for monitoring fatigue damage evolution in Hadfield steel, unlike existing AE fatigue monitoring methodology, which relies solely on the analysis of AE parameter trends.

## 1. Introduction

The railway network is becoming busier than before, with the rolling stock travelling at faster speeds and higher loads being exerted on the rails. The switch and crossing system is an important system used to divert trains from one route to another on the railway network. An important part of the switch and crossing system is the crossing nose, which is subjected to high dynamic impact loads [1]. The most commonly used material to manufacture a crossing nose is Hadfield steel. High-impact loading combined with rolling contact stress and friction might lead to fatigue crack initiation and propagation in a Hadfield manganese steel crossing nose [2,3,4]. The deterioration of a Hadfield steel crossing nose under service is detrimental to passengers’ ride quality; hence, it is vital that a sound maintenance strategy be used for the realisation of high-standard railway service. At present, preventive maintenance is still the main maintenance type for Hadfield steel crossing noses and relies heavily on the personnel’s experience [5]. Currently, the railway industry is focusing more on predictive maintenance using an accurate evaluation of the condition of the assets and less on time-based preventive maintenance. Inspection and monitoring can play a vital role in achieving predictive maintenance for operational switches and crossings. Due to the large scattering generated with the coarse-grained microstructure, the conventional ultrasonic testing used for plain line inspection is difficult to perform on a Hadfield steel crossing nose. Instead, liquid penetrant and visual inspection are two main methods for detecting surface discontinuity in an in-service crossing nose. However, the use of time-consuming inspection techniques is subjected to accessibility and geographical constraints, and by the time a surface discontinuity is detected, the structural condition of the crossing may have been severely weakened, indicating that immediate replacement or repair is needed [6]. Hence, to assess the condition of a Hadfield steel crossing nose more accurately, the use of condition monitoring techniques to complement the existing inspection technologies becomes increasingly necessary. 

Acoustic emission (AE) testing is gaining more popularity as a versatile means for monitoring fatigue crack growth (FCG) behaviour in a variety of materials. It is based on the conversion of elastic waves released from defect growth in a loaded structure into electrical signals using a piezoelectric sensor for either online or post-analysis. There are two fundamental types of analysis for AE signals, namely, waveform-based analysis and parameter-based analysis. Compared with waveform analysis, parameter-based analysis has advantages such as faster data mining and processing [7]. A common AE signal detection technique used in parameter-based AE systems is to compare the signal against a certain threshold. Figure 1 shows a typical AE signal waveform and the commonly used parameters for analysis. Amplitude refers to the highest voltage in the waveform [8]. Count is defined as the number of threshold crossings experienced by the waveform [8]. Duration represents the time between the first threshold crossing and the last threshold crossing [8]. Risetime is defined as the time interval between the first threshold crossing and the signal peak [8]. MARSE Energy is represented as the measure of the area under the envelope of the rectified linear voltage time signal from the sensor [9]. Absolute energy is calculated as the integration of the squared voltage signal divided by a reference resistance over the duration [9]. 

The characteristics of AE parameters during the FCG process have been investigated in many previous studies. Some studies mainly focused on the trend in AE parameters during the FCG process of materials. For instance, the duration of AE signals showed a significant increase of 600–900 cycles before the final failure of the aluminium plate during a tension–tension fatigue test [10]. On the other hand, various studies attempted to establish a correlation between the cumulated release of AE signals and crack length. For instance, during the stable FCG process of A572-G50 steel, Barsoum et al. [11] found that the cumulative energy of AE signals increased in a linear fashion with fatigue cycle number. When the crack became critically active, a rapid increase in cumulative energy occurred. Yu et al. [12] reported an exponential increase in cumulative absolute energy similar to that of crack length during the FCG process of A572G50 steel. Moreover, AE absolute energy rate has proven to be more suitable than the count rate in fatigue life prediction. Chai et al. [13] and Kumar et al. [14] reported that the presence of two sub-stages during the stable FCG process can clearly be revealed using the slope change in cumulative energy versus stress intensity factor range (Δ*K*) plot. Meanwhile, the power-law dependence of AE count rate and energy rate on (Δ*K*) during the stable FCG process was proved for various materials in previous studies. These relationships between AE parameters and Δ*K* have been established in steels, aluminium alloy, and magnesium alloy to make inferences about the remaining fatigue life [10,12,15,16,17]. 

The above studies proved that AE parameters are sensitive to damage progress. However, there are other important aspects of AE measurements, which are based on the distribution parameters of the received signals. An analysis of the AE parameter’s distribution serves as a useful complement to the analysis of trends in the AE parameters. AE signals have been observed to follow the power law probability distribution during the failure process of geological materials. Moreover, the power law exponent is reduced systematically when damage accumulates [18,19,20]. However, there is still a lack of knowledge regarding the evolution of the distribution of AE parameters for metallic material during the fatigue failure process. The accuracy of a damage assessment of structures in service using an analysis of trends in AE parameters cannot be guaranteed due to the potential presence of multiple signal-generation mechanisms, as the parameter trends related to crack growth may be distorted [21]. Moreover, the absolute values of AE parameters are subjected to the influence of external factors including materials and geometric effects, data acquisition settings, etc. [22,23]. Consequently, it is not adequate to evaluate the condition of structures in service solely based on an analysis of AE parameter trends. Compared with parameter trends, characterising the distribution of AE parameters based on the fitted power law has the advantage of being independent of the absolute values of the parameters; hence, it is less affected by the impact of external factors [24]. Therefore, analysis of both AE parameter trends and distributions is highly desirable for accurately evaluating the condition of structures in service. In view of this, the main objectives of this study are twofold, namely, (i) to study the trend in AE signals of Hadfield steel during the FCG process and establish the correlation between AE signals and crack length and (ii) to characterise fatigue damage evolution based on the evolution of the probability distribution of AE parameters using the fitted power law. The novelty of this research lies in the use of the fitted power law distribution of AE parameters for monitoring fatigue damage evolution in Hadfield steel, unlike existing AE fatigue monitoring methodology, which relies solely on an analysis of AE parameter trends. 

The rest of this paper is organised as follows: Section 2 describes the experimental setup and procedures and explains the numerical technique used for determining the power law exponents. Section 3 discusses the experimental results. Section 4 concludes this work.

## 2. Methodology

### 2.1. Materials and Specimens

Standard single-edge notched samples (numbered 1 through 4) of dimensions 120 mm × 20 mm × 10 mm were cut from a Hadfield steel section of UIC 866 grade (1.2 C and 11–14 Mn in wt%). The notch depth and notch root radius were 9 mm and 0.25 mm, respectively, for all samples. The geometry of the samples is shown in Figure 2. The mechanical properties of Hadfield steel are shown in Table 1.

The optical micrograph in Figure 3 shows the microstructure of Hadfield steel. The microstructure was found to contain large austenite grains with some carbide inclusions present at the grain boundaries.

### 2.2. Fatigue Crack Growth Testing

Samples 1 and 2 were tested using three-point bending and samples 3 and 4 using four-point bending. Although rail stresses at the wheel–rail interface involve rolling contact fatigue loads, certain types of cracks away from the contact location, such as vertical cracks in the web or foot of the rail, will grow due to bending stress only [25]. Hence, the bending fatigue tests conducted in the laboratory were partially representative of the loading condition experienced by in-service crossings. The samples were pre-cracked using an Amsler 20 KN Vibrophore electro-mechanical high-frequency fatigue machine. The pre-cracked lengths for samples 1 through 4 were 10.6 mm, 11.25 mm, 9.55 mm, and 10.45 mm, respectively. A sinusoidal wave loading pattern was used during the fatigue tests. The maximum load and load amplitude were set at 3.75 kN and 1.5 kN, respectively. The loading frequency was around 65 Hz. The crack length was monitored using the direct current potential drop (DCPD) technique throughout the experiments, and the fatigue crack growth rate was calculated using the 5-point secant method. The stress intensity factor range Δ*K* is defined as
(1)$$\Delta K=K_{max}-K_{min}$$
where $K_{max}$ and $K_{min}$ are the maximum and minimum stress intensity factors, respectively, calculated for the maximum load $P_{max}$ and the minimum load $P_{min}$ within a loading cycle, respectively. Under current loading configurations, the equations used for calculating Δ*K* are shown below [26,27]:(2)$$\Delta K=\frac{6\Delta P}{B\sqrt{W}}a^{1/2}Y(three\;point\;bending\;fatigue\;test)$$
(3)$$\Delta K=\frac{3\Delta P(\frac{L-Li}{2})}{BW^{3/2}}Y\;(four\;point\;bending\;fatigue\;test)$$
where ΔP = load range; a = crack length; B = thickness of the sample; W = width of the sample; *L* = supporting span; and *L_i_* = loading span. Y is a dimensionless function dependent on the loading configuration and geometry of the sample, and it is calculated as shown in Equations (4) and (5). It should be noted that deviations in the calculated results may exist since the conditions of the present tests were not completely identical to the test conditions where Equations (4) and (5) were derived [26,28].


*Three point bending fatigue test*

(4)
$$Y=\frac{1.99-\left(\frac{a}{W}\right)\left(1-\left(\frac{a}{w}\right)\right)\left(2.15-3.93\left(\frac{a}{w}\right)+2.7{\left(\frac{a}{w}\right)}^{2}\right)}{\left(1+2\left(\frac{a}{w}\right)\right){\left(1-\left(\frac{a}{w}\right)\right)}^{\frac{3}{2}}}$$




*Four point bending fatigue test*

(5)
$$Y=1.99{\left(\frac{a}{w}\right)}^{\frac{1}{2}}-2.47{\left(\frac{a}{w}\right)}^{\frac{3}{2}}+12.97{\left(\frac{a}{w}\right)}^{\frac{5}{2}}-23.17{\left(\frac{a}{w}\right)}^{\frac{7}{2}}+24.87{\left(\frac{a}{w}\right)}^{\frac{9}{2}}$$



### 2.3. Acoustic Emission Technical Features

#### 2.3.1. Acoustic Emission Monitoring

The AE system procured from the Physical Acoustic Corporation (PAC) was used to capture the AE signals generated during the tests. AE signals were captured using two acoustic emission (AE) sensors R50a with operating frequencies of 150–700 kHz, procured from the Physical Acoustic Corporation. The sensors were mounted to the sample surface approximately 20 mm away on either side of the cracked area using duct tape, and Vaseline was used as the couplant between the sensor and the sample surface. The signal recorded with the sensors was amplified using a PAC preamplifier with a gain set at 40 dB. The PAC AEwin v2 software package was used to log AE data. The threshold was set at 40 dB. The peak definition time, hit definition time, and hit lockout time were set at 300 µs, 600 µs, and 1000 µs, respectively. The maximum duration was set at 25,000 µs. The sampling rate was set at 2 MSamples/s. The schematics of experimental setup for three-point and four-point bending fatigue tests are shown in Figure 4.

#### 2.3.2. Power Law Statistics

Power laws are probability distributions of the following form
(6)$$p\left(x\right)\propto \;x^{-\alpha }$$
where α is a constant parameter of the distribution known as the exponent. Unlike other common statistical distributions, such as the Gaussian distribution, exponential distribution, etc., a power law distribution has no typical scale or size for the variable. In practice, the power laws hold only above a certain lower cut-off *x_min_*, and in this case, the tail of the distribution follows a power law. In this study, the Python package powerlaw was used to analyse the distribution of AE parameters. A power law was fitted to the probability distribution of the duration and absolute energy of AE signals using a maximum-likelihood method, and the power law fit spanned at least three bins in all cases. This numerical technique consists of studying the behaviour of the exponent α fitted using the maximum-likelihood method as a function of a varying lower cut-off. Logarithmic binning was used to weaken the impact of the greatly reduced probability of observing large values in the distributions on the reliable estimation of their probabilities of occurrence. The optimal lower cut-off was determined using the one resulting in the minimal Kolmogorov–Smirnov distance between the data and the fit. The maximum likelihood estimators (MLEs) of the exponent α are described as
(7)$$\alpha =1+n\times {\left[\sum_{i=1}^{n}\ln\frac{x_{i}}{x_{min}}\right]}^{-1}$$
where *x_i_*, *i* = 1,2, …, *n* are the observed values of *x* such that *x* ≥ *x_min_*. The standard error *σ* is described as
(8)$$\sigma =\left(\frac{\alpha -1}{\sqrt{n}}\right)+O(\frac{1}{n})$$

## 3. Results and Discussion

### 3.1. Correlation between AE Duration and Crack Length

Figure 5 and Figure 6 show the time history for crack length with duration and cumulative duration for samples 1 through 4. The final failure occurred at 130 s, 97 s, 382 s, and 341 s for samples 1 through 4, respectively, and the crack length at failure occurred at approximately 14.7 mm, 15.4 mm, 15.7 mm, and 16.0 mm, respectively. The relatively intense AE activity at the early stage of the failure process was observed in all samples. This may be related to the gradual release of compressive residual stress generated during the pre-cracking process as the crack propagated. For samples 1 and 2, approximately 20 s before final failure (indicated with the arrow), the moving average value started to increase more sharply. The corresponding cumulative duration also showed a sharp increase at around the same time, hence providing early warning against final failure. Furthermore, the number of AE hits with much higher duration (above 5000 us) increased rapidly when the crack length of both samples was in the range of 14–14.2 mm (indicated with the arrow) at approximately 123 s and 85 s, respectively. The occurrence of AE hits with similar duration values before final failure was also reported in recent studies, and some of them might be related to a shift in the cracking mode from tensile to shear [13,29]. For samples 3 and 4, the moving average showed a sharp increase from approximately 100 s and 143 s (indicated with the arrow and the zoomed portion), respectively. The sharp increase in the moving average lasted around 25 s, and then a slowdown was observed for both samples. The slope of the cumulative duration vs. time curve showed a rapid increase at around the same time. Unlike samples 1 and 2, this rapid increase in cumulative duration occurred much earlier before the final failure, and the slope of the cumulative duration remained generally constant afterwards. This difference may indicate the influence of loading configuration on the AE cumulative trend. For both samples, when the majority of AE hits with a duration above 5000 us, the crack length began to occur in the range of 12–12.5 mm (indicated with the arrow) at approximately 260 s and 180 s, respectively, suggesting the feasibility of using AE hits with high duration as an indicator for imminent failure.

Figure 7 shows a sample after the final failure occurred and micrographs of the crack path. A mixture of both transgranular and intergranular cracking modes was observed for all samples using an optical analysis of the crack path, as shown in Figure 7. Previous studies [30,31] confirmed that AE signals related to fatigue crack growth should concentrate in a banded region, where, in general, the low-amplitude signal has a short duration and the high-amplitude signal has a longer duration. An intergranular fracture is a faster process than a transgranular fracture, as the larger plastic zone associated with transgranular cleavage needs more material to be deformed at any instant, making the transgranular fracture a slower process. The faster fracture process creates more surface area per unit of time, releasing higher energy compared to a slower fracture process and hence, causing AE signals with higher amplitude and duration [32]. Moreover, previous studies have reported that the chance of intergranular fracture generating more intense signals with higher duration increases with (Δ*K*) [33]. Hence, the source mechanism underlying AE hits with high duration in this study can be attributed to intergranular fracture.

### 3.2. Correlation between the AE Duration Rate and Crack Growth Rate

Figure 8 shows the fatigue crack growth rates (*da*/*dN*) and AE duration rates (*dD*/*dN*), calculated using a 5-point secant method, versus ΔK on the log–log scale. The effect of loading configuration on both crack growth rate and duration rate can be clearly observed. For all samples, the results indicate an approximately linear relationship between log(*da*/*dN*) and $\log\left(\Delta K\right),$ which can be described using Paris’s law, as illustrated in Equation (8). However, compared to the four-point bending samples, the three-point bending samples exhibited higher crack growth rates. It was observed that the *dD*/*dN* vs. ΔK plot could be divided into two parts based on time corresponding to the transition point indicated with the arrow, namely, Part 1 and Part 2. The time when the transition occurred agreed well with the time when the sharp increase in the moving average and cumulative values occurred. During Part 1 of the stable FCG process, the compressional residual stress generated during the pre-cracking process was released as the crack propagated and the residual stress decreased with the distance from the pre-crack tip. Hence, the decrease in *dD*/*dN* with Δ*K* in Part 1 may be due to the smaller influence of crack growth on the trend in AE compared with the release of residual stress, given the relatively small crack length. During Part 2 of the stable FCG process, *dD*/*dN* increased sharply with Δ*K* in the three-point bending samples, whereas a relatively slow increase was observed for the four-point bending samples. The sharp increase in *dD*/*dN* with Δ*K* for the three-point bending samples is attributable to the higher work hardening effect of the Hadfield samples due to cyclic loading directly on the ligament part above the cracked area, resulting in a more gradually evolving brittle fracture. It should be noted that *da*/*dN* did not exhibit the same two-stage behaviour as *dD*/*dN*, as it is primarily driven by Δ*K*.

Under both loading configurations, the relationship between *da*/*dN* and Δ*K* was fitted to Paris’s law for all samples:(9)$$\frac{da}{dN}=C\Delta K^{m}$$

Similarly, the relationship between *dD*/*dN* and Δ*K* was fitted to the power law as [14,31]
(10)$$\frac{dD}{dN}=B\Delta K^{p}$$
where *B*, *C*, *m*, and *p* are constants for a particular material and loading condition. The size of the power law *p* exponent indicates how fast the AE parameter value accumulates. Theoretically, *p* equals *m* + 2. However, in real experimental cases, a deviation from the theoretical value always occurs due to the effect of acquisition settings and signal attenuation [31]. Table 2 shows the fitting parameters (*C*, *m*, *B*, and *p*). For samples 1 and 2, m lies in the range of 2–3, whereas for samples 3 and 4, m lies in the range of 1–2. Moreover, although samples 3 and 4 exhibited higher duration values over the majority of the Δ*K* range (29.53 MPa·m^1/2^ ≤ Δ*K* ≤ 52.55 MPa·m^1/2^), samples 1 and 2 exhibited higher *p* values. This indicates that *p* values, namely, the slope of log(*dD*/*dN*) versus log(Δ*K*), are more closely related to *da*/*dN* than *dD*/*dN*. The faster increase in the duration rate in the three-point bending tests can be explained by two factors. Firstly, during three-point bending tests, the crack growth rate increases more rapidly. The duration of crack growth-related signals can be assumed to be closely related to the released energy during the FCG process, and its cumulative sum was assumed to be proportional to the crack growth [31]. Therefore, a faster increase in crack growth rate results in more rapid increase in the duration rate. Secondly, as the cyclic loading is applied directly on the uncracked ligaments above the crack area, the effect of work hardening on crack growth is expected to be higher with a more gradually evolving brittle fracture occurring in three-point bending samples, hence, resulting in a more rapid accumulation of AE duration [34]. For an in-service crossing nose, the change in actual load condition occurs due to wear on the crossing nose and to wear from wheel profiles of passing vehicles. Therefore, it is necessary to determine the associated loading conditions for more accurate data interpretation, given the obvious effect of loading configuration on the AE rate shown above.

### 3.3. Power Law Statistics

Figure 9 and Figure 10 show the probability distribution and the corresponding power law fit for the duration and energy of the samples during the stable FCG process. The optimal lower cut-off during Part 1 of the stable FCG process was determined using the cut-off that resulted in the minimal Kolmogorov–Smirnov distance between the data and fit. For comparison purposes, the same range of data during Part 2 of the stable FCG process was used to obtain the probability density function and the corresponding power law fit. The fluctuation in the probability distribution plots may be due to the combined effect of finite data size and the mixing of different AE source mechanisms. Although the power law did not produce perfect fits, the exponents for both absolute energy and duration during Part 2 were smaller than those during Part 1 in all samples, which could be interpreted in the sense that the relative percentage of AE signals with higher energy and duration increased during Part 2. The decrease in power law exponents as damage accumulated is consistent with previous studies [35,36]. 

The energy exponent for dislocation-related acoustic emission signals for 304 L austenitic stainless steel was reported to be 2.14 in a previous study, and the measured energy exponents should not be far from that value, given the predominantly austenitic microstructure of Hadfield and the occurrence of severe dislocation activities during the FCG process [2]. As shown in Figure 9 and Figure 10, the energy exponents α for samples 1 through 4 during Part 2 were 2.12 ± 0.06, 1.98 ± 0.04, 1.98 ± 0.01, and 2.27 ± 0.01, respectively, and the values were indeed not far from the one in austenitic stainless steel.

Table 3 summarises the mean values of the power law exponents for the three- and four-point bending samples. The mean values of exponents for the three-point bending samples were lower than those for the four-point bending samples. Given the higher crack growth exponent in the three-point bending samples, this suggested a possible correlation between the crack growth exponent and the fitted power law exponent. The above results clearly show that analysis of the fitted power law distribution of AE parameters can be carried out in conjunction with a conventional analysis based on parameter trends to provide a new approach for evaluating the condition of a Hadfield steel crossing nose.

## 4. Conclusions

The AE characteristics of Hadfield steels during three- and four-point bending fatigue tests were investigated in this study. For the first time, the effectiveness of analysing the distribution of AE parameters based on a fitted power law for monitoring the evolution of fatigue damage in metallic material was demonstrated. AE characteristics under different loading configurations were compared, thus providing insight into the effect of loading configuration on evolutionary trends in AE parameters.

The conclusions drawn from this study are as follows:(1)A sharp increase in the moving average and cumulative sum of AE duration occurs before the final fatigue failure.(2)The occurrence of AE signals with specific features occurs when the crack length is within a certain range. AE hits with high duration (above 5000 us) begin to occur when the crack length is in the range of 14–14.2 mm and 12.2–12.5 mm for samples subjected to three- and four-point bending fatigue, respectively.(3)Both fatigue and AE behaviours can be affected by the loading configuration. Compared with four-point bending fatigue tests, cracks grew more rapidly in the three-point bending fatigue tests. Under both three- and four-point bending configurations, *dD*/*dN* decreased with delta *K* until a transition during Part 1 of the stable FCG process and then exhibited an increasing trend during Part 2 of the stable FCG process. However, during Part 2, the duration rate increased sharply with Δ*K* in the three-point bending tests, whereas a relatively slow increase was observed in the four-point bending tests.(4)Although AE absolute energy and duration values were not fitted well with the power law, the fitted exponents based on the maximum likelihood method for both parameters during Part 2 of the stable FCG process were smaller than those during Part 1 in all cases.(5)AE is a promising complement to the existing methods for qualitative evaluation of the condition of a Hadfield steel crossing nose, and it should be further investigated for quantitative characterisation of damage in Hadfield steel crossing noses.

The results and conclusions presented above clearly indicate that a more accurate assessment of the condition of a Hadfield steel crossing nose is achievable using the AE technique. In an actual operating railway environment, AE sensors need to be attached to the sides of a crossing nose. The system is triggered to acquire AE signals once the rolling stock approaches a crossing nose. AE signals acquired from a crossing nose in good condition can serve as the benchmark. The condition of a Hadfield steel crossing nose can then be evaluated automatically using a combined analysis of trends and fitted power law distributions in AE parameters, which can help engineers optimise maintenance and renewal planning.

## Figures and Tables

**Figure 1 sensors-23-06561-f001:**
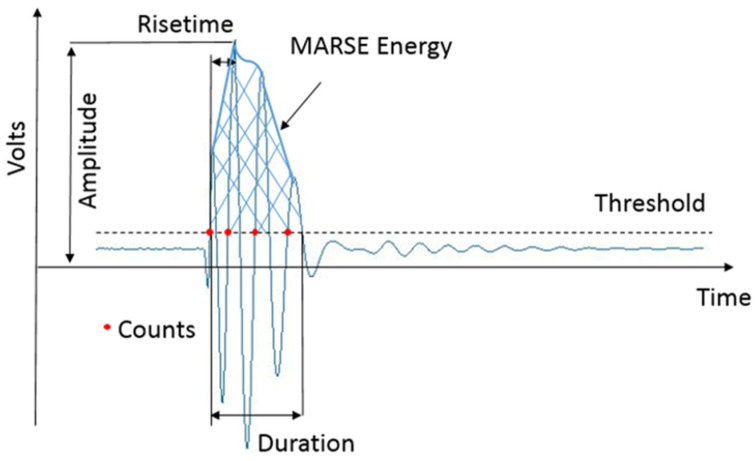
AE signal and the commonly used parameters.

**Figure 2 sensors-23-06561-f002:**
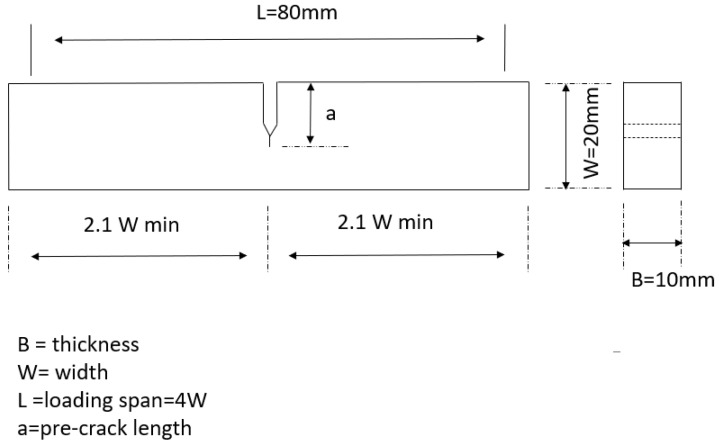
Schematic diagram showing the geometry of the samples used for fatigue crack growth tests.

**Figure 3 sensors-23-06561-f003:**
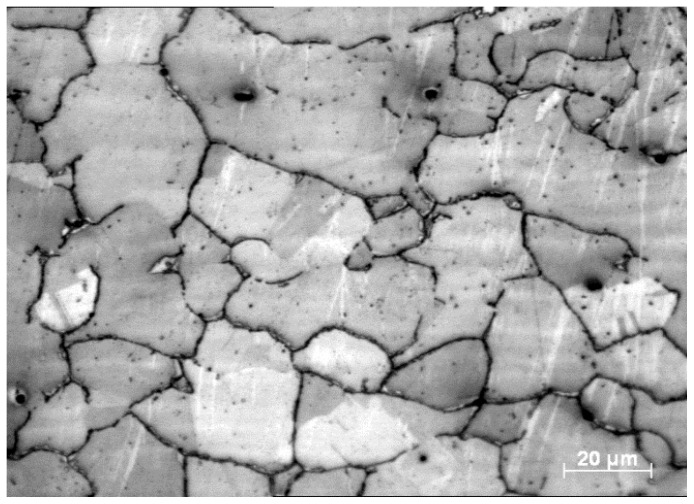
Microstructure of Hadfield steel.

**Figure 4 sensors-23-06561-f004:**
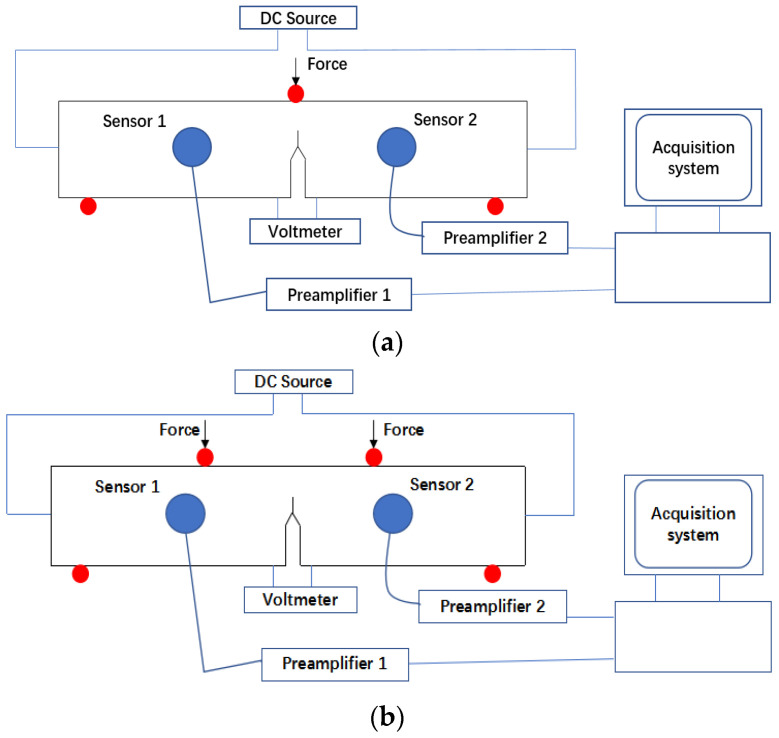
Schematic showing the experimental setup for the (**a**) three-point bending fatigue test (**b**) four-point bending fatigue test.

**Figure 5 sensors-23-06561-f005:**
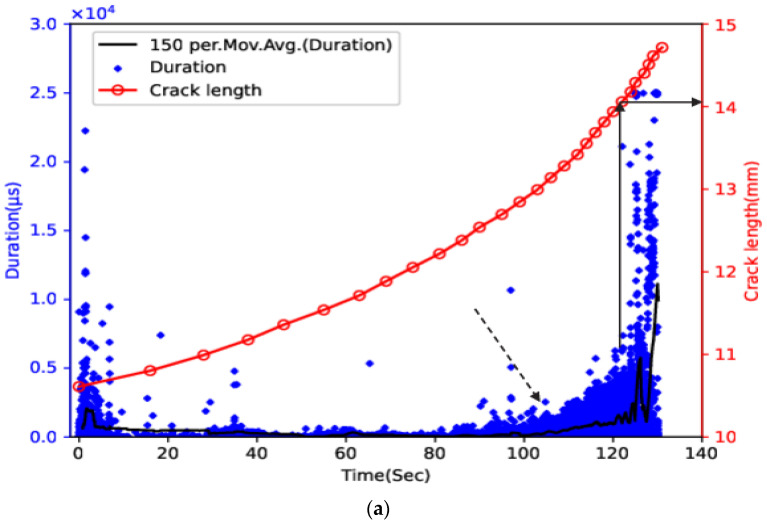
Time history for crack length and (**a**) duration and its moving average and (**b**) cumulative duration for sample 1 and (**c**) duration and its moving average and (**d**) cumulative duration for sample 2.

**Figure 6 sensors-23-06561-f006:**
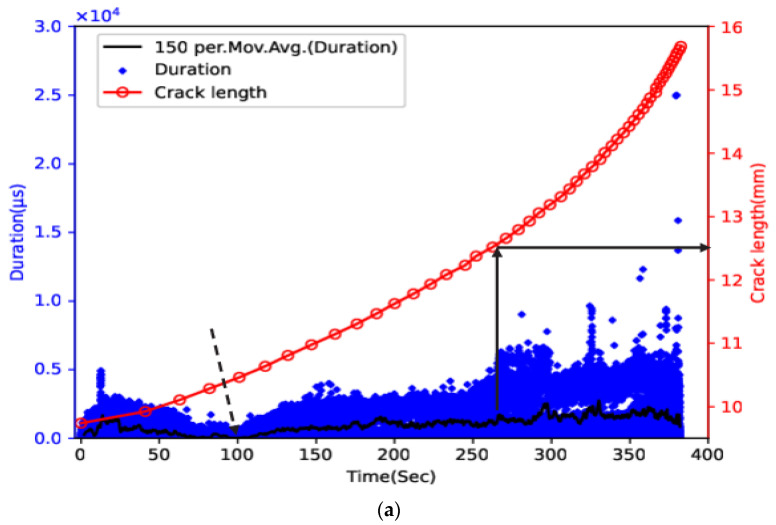
Time history of crack length and (**a**) duration and its moving average and (**b**) cumulative duration for sample 3 and (**c**) duration and its moving average and (**d**) cumulative duration for sample 4.

**Figure 7 sensors-23-06561-f007:**
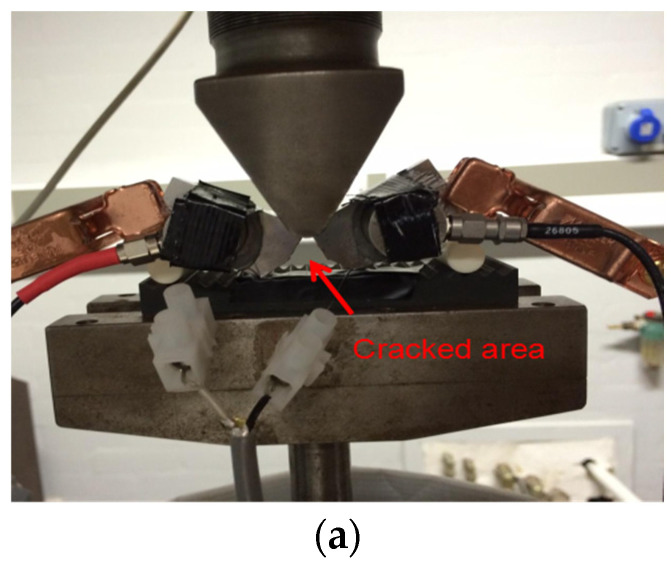
(**a**) Photograph showing a sample after the final failure occurred. (**b**) Micrographs showing the crack path after a fatigue crack test indicative of a mixture of both transgranular and intergranular cracking.

**Figure 8 sensors-23-06561-f008:**
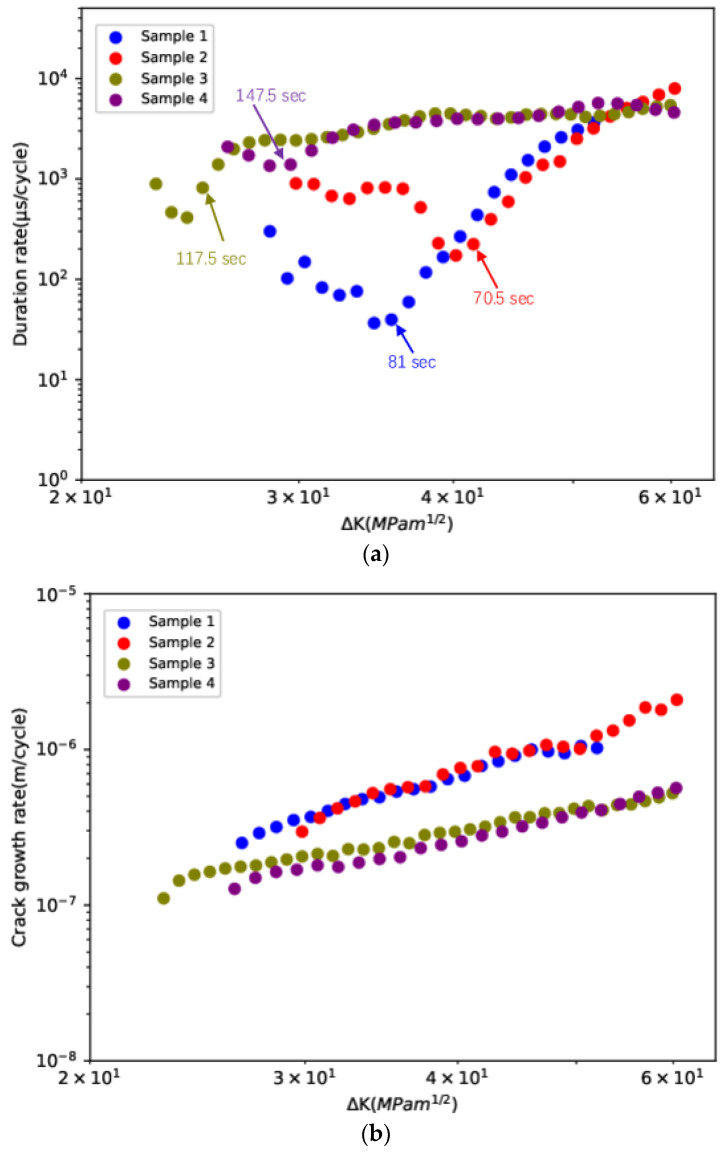
(**a**) AE duration rates (*dD*/*dN*) and (**b**) fatigue crack growth rates (*da*/*dN*) versus Δ*K* on the log–log scale.

**Figure 9 sensors-23-06561-f009:**
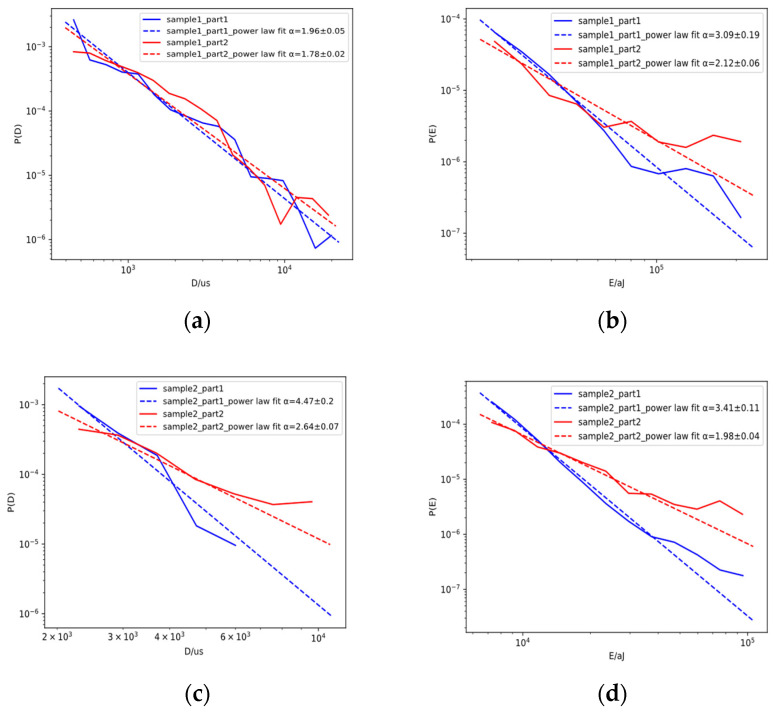
Probability distribution and the corresponding power law fit for (**a**) duration and (**b**) energy for sample 1 and (**c**) duration and (**d**) energy for sample 2 during Part 1 and Part 2 of the stable FCG process.

**Figure 10 sensors-23-06561-f010:**
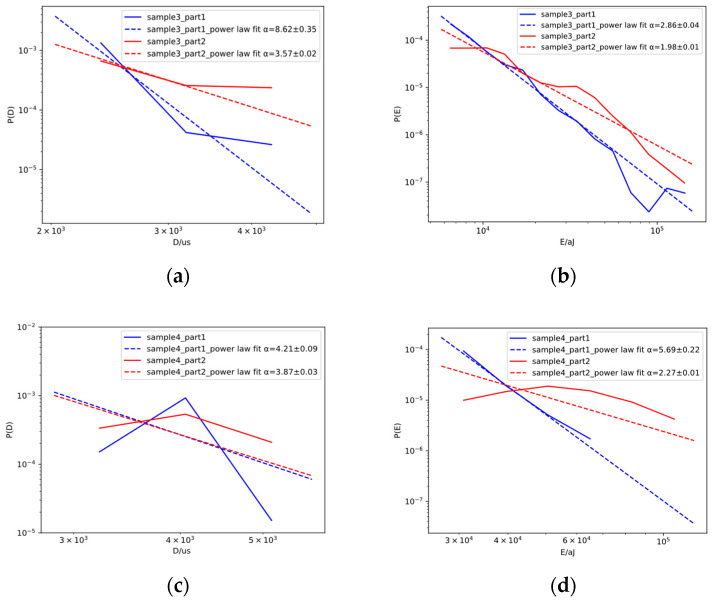
Probability distribution and the corresponding power law fit for (**a**) duration ad (**b**) energy for sample 3 and (**c**) duration and (**d**) energy for sample 4 during Part 1 and Part 2 of the stable FCG process.

**Table 1 sensors-23-06561-t001:** Mechanical properties of Hadfield steel.

Brinell Hardness	Tensile Strength	Yield Strength	Elongation at Break
200 HB	880 MPa	320 MPa	20%

**Table 2 sensors-23-06561-t002:** Fitting values for crack growth and AE constants for Hadfield steel.

	Sample ID	Pre-Crack Length/mm	*m*	*C*	*p*	*B*
Three-point bending fatigue	1	10.6	2.1226	3 × 10^−10^	7.2335	2 × 10^−9^
	2	11.25	2.4203	9 × 10^−11^	3.6365	0.0013
Four-point bending fatigue	3	9.55	1.5355	3 × 10^−9^	2.1288	5.39
	4	10.45	1.9011	9 × 10^−10^	1.7407	17.903

**Table 3 sensors-23-06561-t003:** Mean values of power law exponents.

	Duration Exponent		Energy Exponent	
	Part1	Part2	Part1	Part2
**Three-point bending sample**	3.22 ± 0.25	2.21 ± 0.09	3.25 ± 0.3	2.05 ± 0.1
**Four-point bending sample**	6.42 ± 0.44	3.72 ± 0.05	4.28 ± 0.26	2.13 ± 0.02

## Data Availability

All data are contained within the article.

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
