# Peer review of "Acoustic Emission Monitoring of Fatigue Crack Growth in Hadfield Steel"

_sensors, 2023, doi:10.3390/s23146561_

Round 1
Reviewer 1 Report
It is interesting to focus on the distribution of AE parameters. The first part of the paper is clear. However, there is room for improvement in presenting the results. Nevertheless, the paper demonstrates the potential of AE in addressing fatigue issues, and it is generally desirable for publication.
Looking forward to seeing deeper insights into the relationship between AE parameter distribution and fatigue in your future work.
1. Why did the authors choose to focus on duration instead of frequency for their analysis? Since the duration can vary depending on the threshold settings, wouldn't it be more appropriate to utilize frequency instead?
2. L276: I couldn't understand why AEs with high durations could be attributed to intergranular fracture. It would be helpful to provide a more detailed explanation in the text.
3. There is an error on line 282. Please correct it.
4. Figure 8 is difficult to read. It would be better to split it into two plots, one for the upper part and one for the lower part.
Accept after minor revision
Reviewer 2 Report
In this study, the authors investigate the correlation of acoustic emission and fatigue crack growth of crossing nose made of Hadfield steel during bending fatigue tests. The paper is of quite good quality, so it could be recommended for publication.
Though the authors have stated that power law was fitted to the probability distribution of duration and absolute energy of AE signals using a maximum-likelihood method, more details related to the fitting of the power law and comparison with other laws is needed.
It is not fully clear what is the main scientific novelty of the paper? This issue should be obviously specified in the abstract and introduction.
Reviewer 3 Report
Dear authors, I wish you success in the field of acoustic monitoring. Pay special attention to the last remark.
